# The Influence of Thermomechanical Conditions on the Hot Ductility of Continuously Cast Microalloyed Steels

**DOI:** 10.3390/ma17184551

**Published:** 2024-09-16

**Authors:** Saham Sadat Sharifi, Saeid Bakhtiari, Esmaeil Shahryari, Christof Sommitsch, Maria Cecilia Poletti

**Affiliations:** 1Institute of Materials Science, Joining and Forming at Graz University of Technology, Kopernikusgasse 24/I, 8010 Graz, Austria; saeid.bakhtiari@k1-met.com (S.B.); esmaeil.shahryari@tugraz.at (E.S.); christof.sommitsch@tugraz.at (C.S.); cecilia.poletti@tugraz.at (M.C.P.); 2K1-MET GmbH, 4020 Linz, Austria; 3Christian Doppler Laboratory for Design of High-Performance Alloys by Thermomechanical Processing, Kopernikusgasse 24, 8010 Graz, Austria

**Keywords:** steels, continuous casting, plastic deformation, dynamic restoration, phase transformation, hot ductility

## Abstract

Continuous casting is the most common method for producing steel into semi-finished shapes like billets or slabs. Throughout this process, steel experiences mechanical and thermal stresses, which influence its mechanical properties. During continuous casting, decreased formability in steel components leads to crack formation and failure. One reason for this phenomenon is the appearance of the soft ferrite phase during cooling. However, it is unclear under which conditions this ferrite is detrimental to the formability. In the present research, we investigated what microstructural changes decrease the formability of microalloyed steels during continuous casting. We studied the hot compression behaviour of microalloyed steel over temperatures ranging from 650 °C to 1100 °C and strain rates of 0.1 $s^{-1}$ to 0.001 $s^{-1}$ using a Gleeble 3800^®^ (Dynamic Systems Inc, Poestenkill, NY, USA) device. We examined microstructural changes at various deformation conditions using microscopy. Furthermore, we implemented a physically-based model to describe the deformation of austenite and ferrite. The model describes the work hardening and dynamic restoration mechanisms, i.e., discontinuous dynamic recrystallisation in austenite and dynamic recovery in ferrite and austenite. The model considers the stress, strain, and strain rate distribution between phases by describing the dynamic phase transformation during the deformation in iso-work conditions. Increasing the strain rate below the transformation temperature improves hot ductility by reducing dynamic recovery and strain concentration in ferrite. Due to limited grain boundary sliding, the hot ductility improves at lower temperatures (<750 °C). In the single-phase domain, dynamic recrystallisation improves the hot ductility provided that fracture occurs at strains in which dynamic recrystallisation advances. However, at very low strain rates, the ductility decreases due to prolonged time for grain boundary sliding and crack propagation.

## 1. Introduction

Steels are mainly produced through a continuous casting process [1]. Continuous casting involves the uninterrupted casting of molten metal into semi-finished forms, such as billets or slabs. This process introduces various mechanical and thermal stresses, such as the forces between the mould and shell, the ferrostatic pressure, i.e., the pressure exerted by the molten steel due to its weight, and the bending force. Therefore, the complexity of this process incorporates several interacting phenomena that significantly influence the mechanical properties, for instance, the formability of the slab [1,2]. Reduced formability of steel slabs during continuous casting enhances crack initiation and leads to their failure. In continuous casting of steels, the term “second ductility minimum” typically refers to a reduction in area (RA) observed during the cooling process after the solidification [1,3]. Depending on the chemical composition of the alloy, the second ductility minimum commonly occurs between 700 °C and 900 °C [2,3].

Figure 1a shows the hot ductility behaviour of the studied microalloyed steel. The data from the hot tensile test are taken from previous work on the same material [4]. As temperature decreases from 1100 °C to 750 °C, the hot ductility of the specimen decreases. Below transformation temperature, ferrite film forms at austenite grain boundaries. The strain concentrates at the soft ferrite phase, leading to the formation of microvoids. The microvoids grow and coalesce, provoking cracks and failure. Several factors influence this loss in ductility during manufacturing, including the nucleation of a soft phase, ferrite, within specific temperature ranges, the presence of stress-concentrating precipitates, and microstructural modifications due to elevated temperatures and applied stresses [5]. 

Beal et al. [6] reported that between 750 °C and 800 °C the formation of thin ferrite film along the austenite grain boundary reduces the ductility, while ductility improves at temperatures lower than 700 °C. Strain concentrates at ferrite as it has lower strength compared to austenite, making the material more prone to failure [6,7]. Moreover, the ductility improvement at high temperatures is attributed to the absence of ferrite and the annihilation of dislocations, which are preferred nucleation sites for precipitates [4,5,8]. Wang et al. [9] studied the influence of the precipitates at grain boundaries in Ti-Mo microalloyed steels on the hot deformation. They showed that NbC precipitates decrease the hot ductility by forming precipitate-free zones (PFZs). Moreover, they reported that irrespective of the deformation conditions, coarse (Ti, Mo)C precipitates formed at grain boundaries, while fine precipitates were uniformly dispersed within the grains. Studies show that the size and distribution of precipitates influence hot ductility. For instance, fine precipitates randomly distributed at the grain boundaries significantly reduce the ductility compared to coarse precipitates evenly distributed within the grains [2].

During hot deformation, microstructure evolves not only due to phase transformations and precipitation but also through dynamic restoration mechanisms. These microstructural modifications, due to dynamic phenomena during deformation, change the mechanical properties and so the ductility [10,11]. Several studies investigated the influence of temperature and strain rate on the ferrite content and hot ductility in continuously casted steels [4,6,7,10,12]. Thermomechanical conditions such as temperature and strain rate influence the formation of ferrite films and precipitates as well as the restoration mechanisms in both phases, ferrite and austenite. Although the interaction between phase transformation and restoration mechanisms is well documented, a comprehensive understanding of their combined effects remains limited.

The concurrent phase transformation changes phase fractions, which in turn impact the strain rate in each phase. The varying strain rate influences the restoration behaviour within phases, modifying strain distribution. However, confirming these hypotheses through experimental research alone is challenging due to difficulties in microanalysing individual phases concerning strain rate, strain, and dislocation density quantification. Mesoscale modelling enables us to understand the underlying phenomena occurring during deformation in each phase and their contribution to the overall deformation behaviour of the material.

The aim of this study is to investigate the microstructural modifications and corresponding restoration mechanisms in microalloyed steel during hot deformation using mesoscale models. This work describes the metallurgical changes occurring during hot compression in the absence of damage and aims to correlate the findings with the reduction in ductility observed in steel products during hot tensile testing as a representation of continuous casting conditions. Our analysis considers the microstructural changes in phase transformation and dynamic restoration mechanisms. We carried out experiments (see Section 2) and developed a mesoscale model (described in Section 3) to achieve our goal.

**Figure 1 materials-17-04551-f001:**
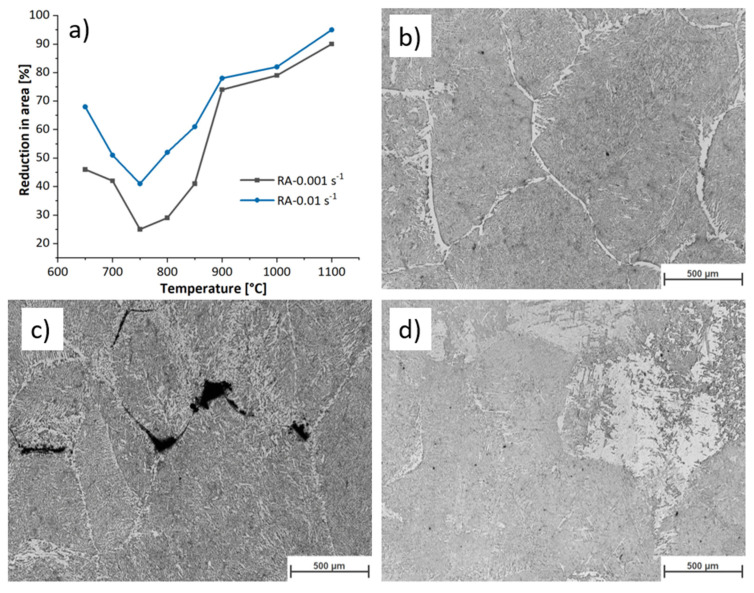
(**a**) Reduction in area of the samples deformed during hot tensile tests at various conditions, and OM images of the deformed sample under strain rate of 0.01 $s^{-1}$ and temperature of (**b**) 650 °C, (**c**) 750 °C, and (**d**) 850 °C. Modified after [13].

## 2. Materials and Methods

### 2.1. Material

Table 1 shows the chemical composition of the studied microalloyed steel. The samples were taken from the slab produced through continuous casting, with the axis aligned parallel to the rolling direction. We heat-treated three specimens at 1200 °C for 360 s, subsequently cooled to 750 °C, 900 °C, and 1100 °C at a cooling rate of 1 °C/s, and then water quenched to obtain the initial microstructure of the alloy before straining.

### 2.2. Ductility Minimum

The hot tensile tests were conducted using a BETA 250-5 thermomechanical simulator (Mechanical Testing Systems, Berlin, Germany) in a vacuum atmosphere. The temperature was controlled by a Pt/Pt-Rh thermocouple welded to the surface at the middle of the specimen’s gauge length. The samples were heated up to the melting temperature using an induction coil. In the BETA 250-5 setup, the induction coil is connected to the machine’s upper part, moving upwards at half speed during the deformation step to concentrate the heating in the centre of the specimen. The specimens were heated to 1450 °C, melting the sample’s interior. The specimens were held at this temperature for 90 s, then cooled to 1250 °C/s at a cooling rate of 5 °C/s. The cooling rate from 1250 °C to the deformation temperatures (650, 700, 750, 800, 850, 900, 950, 1000, and 1100 °C) was 1 °C/s. The specimens were held for 10 s before deformation began. At the deformation temperature (Td), the hot tensile tests were performed at strain rates of 0.01 $s^{-1}$ and 0.001 $s^{-1}$. After rupture, the area of the fractured surface was optically measured. The details of the experiments and the measurements can be found in [4]. The reduction in area RA (%) is given by Equation (1).
(1)$$RA\left(\%\right)=\left(\frac{A_{0}-A_{f}}{A_{0}}\right)\times 100$$
where $A_{0}$ and $A_{f}$ correspond to the initial cross-section and the measured fracture surface, respectively.

### 2.3. Hot Compression Tests

Since the tensile tests provide flow curves up to small plastic strains due to necking, we conducted hot compression tests to obtain information on the flow stresses and the microstructure up to larger strains. The selected device for this is a Gleeble^®^ 3800 thermomechanical simulator. We followed the thermomechanical route similar to the tensile test described in Section 2.2 without the melting step, as illustrated in Figure 2a. Figure 2b depicts the dimension of the compression specimen. The temperature was controlled by a S-type thermocouple (thermocouple 1) welded in the middle of the sample. Another S-type thermocouple (thermocouple 2) welded at the sample’s edge measures the temperature gradient along the deformation axis. The samples were compressed in an Ar-protective atmosphere, followed by immediate water quenching to room temperature (RT) to preserve the deformed microstructures. We performed each test condition at least twice to ensure the reproducibility of the experiments. Table 2 depicts the window of the experimental parameters. To track the influence of the deformation on the microstructure, we conducted hot compression on various final strains at 900 °C and 1100 °C, as detailed in Table 2.

The elastic strain was removed from the stress-strain curves to obtain the flow curves, and the data were smoothed and sampled for a strain step of 0.001 using OriginLab 2020 software.

### 2.4. Metallography

We sectioned the compressed specimens longitudinally (i.e., along the load axis) for microstructural analysis. The cut surface is mechanically ground with SiC papers up to P1200 and polished in two steps. First, for 5 min with an alumina suspension with 1 µm of particle size and then for 15 min with a colloidal silica suspension. We used 3% Nital etchant to identify the ferrite phase and austenite, which transforms to martensite, and CRIDA-QT Plus etchant (which is a picric acid-based solution) from CRIDA Chemie manufacturer for 8–12 min to identify prior austenite grain boundaries.

LOM Zeiss Axio Observer Inverted (Carl Zeiss AG, Oberkochen, Germany) is then used to image the microstructures. We used ImageJ 1.53e software to identify and quantify the ferritic phase from the transformed austenite. We converted the optical microscope (OM) images to 8 bits (i.e., black and white tones) to differentiate ferrite and martensite. We also determined the average grain using this software using the Heyn Lineal Intercept method from the ASTM E112 standard [14]. Each measurement averages five horizontal and five vertical lines in each image. Measurements were repeated on three images for each deformation condition. 

Figure 3 shows the initial microstructures of three representative samples. The measured average grain sizes for 750 °C, 900 °C, and 1100 °C are 145.2 µm, 151.8 µm, and 155.1 µm, respectively. The measured values have an average standard deviation of 14.2 µm. The austenitisation temperature and time control the prior austenite grain size. Therefore, for a given time, the austenite grain size does not vary significantly with the austenitisation temperature. We considered approximately 150 µm the average initial austenite grain size before deformation. The initial grain size of the hot tensile test was coarser (approximately 50% coarser) than what we produced in the current study due to heating at a higher austenitisation temperature.

## 3. Model Setup

The developed model describes the evolutions of the microstructure and the flow stress of steels deformed in the single- and two-phase fields. Our model consists of constitutive equations to correlate the stress with the microstructure, strain rate partitioning between the two phases, and rate equations to account for the evolution of dislocation densities. We considered the phase evolution using a power law equation validated with experimental results (see Section 3.2).

### 3.1. Yield Stress

The yield stress $\sigma_{y}$ is modelled phenomenologically using the approach proposed in [15] as read in Equation (2):(2)$$\sigma_{y,x}=\frac{1}{a_{y,x}}ln\left\{{\left(\frac{Z}{A}\right)}^{\frac{1}{n_{y,x}}}+{\left[{\left(\frac{Z}{A}\right)}^{\frac{2}{n_{y,x}}}+1\right]}^{0.5}\right\},\;\;\;\;\;\;\;\;\;\;x=\alpha +\gamma \;,\;\gamma$$
where subscript x denotes the austenite (γ) and ferrite and austenite (α+γ) domains, and the Zener-Hollomon (*Z*) parameter [8] correlates the temperature, *T*, and the strain rate, $\dot{\varepsilon }$ as expressed in Equation (3):(3)$$Z=\dot{\varepsilon }exp\left(\frac{Q_{y,x}}{RT}\right),\;\;\;\;\;\;\;\;\;\;x=\alpha +\gamma \;,\;\gamma$$

The activation energy, $Q_{y}$, and parameters A, $\alpha_{y}$, $n_{y}$ for both domains were calculated and listed in Appendix B in Table A2. R is the universal constant of gases. 

### 3.2. Phase Transformation Model

Following the thermomechanical processing route illustrated in Figure 2, the amount of ferrite withing the microstructure changes in two stages: (i) during continuous cooling from austenitisation temperature, i.e., 1200 °C, to the deformation temperatures, and (ii) during isothermal deformation. We calculated the volume fraction of ferrite during continuous cooling using CCT diagrams. In the second stage, austenite transforms during isothermal deformation. At different strain rates, the nucleation time varies, resulting in different phases. We calculated the phase evolution during deformation using TTT diagrams at each temperature. CCT and TTT diagrams were generated using JMatPro v14 software [16]. The total volume fraction of ferrite is the sum of the volume fraction obtained during cooling, followed by the ferrite formed isothermally. We fitted the ferrite volume fraction obtained from CCT and TTT diagrams to an allometric function of strain using OriginLab 2020 software (Equation (4)). Finally, the ferrite fraction ($f_{\alpha }$) at each deformation step is correlated to the respective deformation temperature and strain rate through Equations (5) and (6).
(4)$$f_{\alpha }=a\varepsilon^{b}$$
where a and b are model parameters correlated to the thermomechanical conditions:(5)$$a=\xi T^{m_{a}}{\dot{\varepsilon }}^{n_{a}},$$
(6)$$b=\psi T^{m_{b}}{\dot{\varepsilon }}^{n_{b}},$$
where ξ, $m_{a}$, $n_{a}$, ψ, $m_{b}$, $n_{b}$ are phase transformation fitting parameters listed in Table 3.

### 3.3. Microstructure Modelling and Dislocation Density

We developed the basis of the current proposed model from the work of Kocks-Mecking on dynamic recovery (DRV) and added concepts of the discontinuous dynamic recrystallisation (dDRX) mechanism [1,17]. In the context of this study, DRX refers to dDRX. The microstructure has two phases: a high stacking fault energy (SFE) phase, ferrite, and a low SFE phase, austenite. The total dislocation density, $\rho_{m}$, in each phase is the sum of immobile dislocation density, $\rho_{i}$, and mobile dislocation density, $\rho_{m}$ following Equation (7):(7)$$\rho_{t,x}=\rho_{i,x}+\rho_{m,x}\;,\;\;\;\;\;\;\;\;\;\;x=\alpha \;,\;\gamma$$

The immobile dislocations pin the mobile dislocations on their glide path, contributing to work-hardening. In contrast, mobile dislocations can freely glide; therefore, their contribution to work-hardening is trivial.

### 3.4. Constitutive Equations

In the current model, the flow stress is composed of two main contributions for each phase: (i) the athermal stress for the long-range interaction between mobile and immobile dislocations through their elastic field, $\sigma_{ath}$, [18], and (ii) the thermal stress $\sigma_{th}$ required to move the mobile dislocations (Equation (8)).
(8)$$\sigma_{x}=M_{x}\left(\tau_{ath,x}+\tau_{th,x}\right),\;\;\;\;\;\;\;\;\;\;x=\alpha ,\gamma$$
where $M_{x}$ is the Taylor factor of each phase. The athermal stress is calculated according to Equation (9) (Taylor equation):(9)$$\tau_{ath,x}=a_{x}^{\prime }\mu_{x}b_{x}\sqrt{\rho_{i,x}+\rho_{m,x}}\;\;,\;\;\;\;\;\;\;\;\;\;x=\alpha ,\gamma$$
where $a_{x}^{\prime }$ is the Taylor constant equal to 0.1 for both phases, $b_{x}$ is the Burgers vector, and $\mu_{x}$ is the shear modulus.

The thermal stress has a fixed value for a given temperature and strain rate and is an output of the model. The upper limit of thermal stress for each phase is defined according to the yield stress $\sigma_{ys,x}$ for each phase at a given temperature and strain rate (Equation (10)):(10)$$\sigma_{th,x}^{0}=\sigma_{ys,x}-\sigma_{ath,x}^{0},\;\;\;\;\;\;\;\;\;\;x=\alpha ,\gamma$$
where $\sigma_{ath,x}^{0}$ is the stress caused by the initial dislocation density. The initial dislocation density at a given deformation condition is correlated to the material yield stress using Equation (11). The yield stress is obtained by Equation (2).
(11)$$\rho_{0,x}={\left(\frac{\sigma_{y,x}}{M_{x}a_{x}^{\prime }\mu_{x}b_{x}}\right)}^{2},\;\;\;\;\;\;\;\;\;\;x=\alpha ,\gamma$$

### 3.5. Dislocation Density Rate

The variation of immobile dislocation density over time for each phase is given by the Kocks-Mecking formalism [19] (Equation (12)):(12)$$\frac{\partial \rho_{t,x}}{\partial t}=\dot{\varepsilon }\left[h_{1,x}\sqrt{\rho_{i,x}}-h_{2,x}\left(\rho_{i,x}\right)\right]\;,\;\;\;\;\;\;\;\;\;\;x=\alpha ,\gamma$$

$h_{1,x}$ and $h_{2,x}$ are the work hardening and the dynamic recovery coefficients of each phase, respectively, and are correlated to the thermomechanical process condition through Equations (13) and (14):(13)$$h_{1,x}=h_{01,x}{\dot{\varepsilon }}^{m_{h1,x}}exp(\frac{m_{h1,x}.Q_{h1,x}}{RT})\;,\;\;\;\;\;\;\;\;\;\;x=\alpha ,\gamma$$
(14)$$h_{2,x}=h_{02,x}{\dot{\varepsilon }}^{{-m}_{h2,x}}exp(\frac{{-m}_{h2,x}.Q_{h2,x}}{RT})\;,\;\;\;\;\;\;\;\;\;\;x=\alpha ,\gamma$$
here $h_{01,x}$, $m_{h1,x}$, $Q_{h1,x}$, $h_{02,x}$, $m_{h2,x}$, and $Q_{h2,x}$ are materials parameters. Table A3 in Appendix B gives the values of these parameters obtained by fitting the experimental flow curves. We assumed that the mobile dislocation density remains constant at a given strain rate during deformation.

### 3.6. Discontinuous Dynamic Recrystallisation (dDRX) 

The dynamic softening caused by dDRX involves two distinct steps: the nucleation of new grains and the growth of those nuclei.

#### 3.6.1. Nucleation

During the plastic deformation of austenite, dislocation density increases until it reaches a critical value for the onset of DRX identified as the critical strain, $\varepsilon_{cr}$ [20,21]. Once the grains reach the critical dislocation density, the nucleation occurs at prior austenite grain boundaries.

This critical strain correlates to the experimental strain at the peak stress εp as $\varepsilon_{cr}=B\varepsilon_{p}$ (B is a constant between 0.5 and 0.8) [20]. Once the strain reaches the critical value, the nucleation starts. Our model assumes a constant nucleation rate, correlating the deformation conditions with the grain boundary energy. The formulation is adapted from [22] as follows (Equation (15)):(15)$$\dot{N}={\left(\frac{\dot{\varepsilon }}{{\dot{\varepsilon }}_{ref}}\right)}^{c_{1}}N_{0}\exp\left(\frac{-\gamma_{gb}b^{2}}{k_{B}T}\right)$$
where ${\dot{\varepsilon }}_{ref}$ is a reference strain rate, $\gamma_{gb}$ is the grain boundary energy, $k_{B}$ is the Boltzmann constant, T is the temperature in Kelvin, and $c_{1}$ and $N_{0}$ are model constants with values listed in Appendix B Table A1.

#### 3.6.2. Growth Model

The difference in the dislocation density between deformed and recrystallised grains provides the driving force for recrystallised grains to grow. The growth rate of the recrystallised grains is a product of the mobility of the grain boundary, $M_{gb}$ with the total pressure, P, exerted on the grain boundary. The mobility of the grain boundary, $M_{gb}$, reads (Equation (16)):(16)$$M_{gb}=M_{0}\exp\left(\frac{{-Q}_{gb}}{k_{B}T}\right)$$
where $Q_{gb}$ is the activation energy for grain boundary movement and $M_{0}$ is the pre-exponential mobility factor given in Appendix B, Table A1.

The grain boundary experiences different pressures, i.e., a positive pressure due to the difference in dislocation density on both sides, a positive capillarity pressure due to grain boundary curvature, and a negative pressure caused by the presence of precipitates at grain boundaries. The microalloyed steel of this work contains no precipitates. During deformation, the capillarity pressure reading ($1.5\gamma_{gb}/{Փ}_{drx}$) where ${Փ}_{drx}$ is the size of the recrystallised grain, is negligible. Therefore, we considered only the pressure caused by dislocation as the main driving force for growth. The stored energy provided by dislocation density reads (Equation (17)):(17)$$P_{gb}=\alpha \mu b^{2}\left(\rho_{i}+\rho_{m}\right)$$

The growth rate of the recrystallised grains is a modified approach proposed by Deschamps et al. [23]. The growth rate of the recrystallised grain ($\dot{Փ}=\frac{d{Փ}_{drx}}{dt}$) is the difference between the velocity of the deformed grain boundary, $v_{gb}$, and the velocity of the recrystallised grain boundary, $v_{drx}$ [23]. This indicates that the velocity of the recrystallised grain reduces the velocity of the deforming grain by nucleating new grains over time (Equation (18)):(18)$$v_{drx}=v_{gb}+\left(\frac{d{Փ}_{drx}}{dt}\right)$$
where $d{Փ}_{drx}$ is the change in the size of recrystallised grain after nucleation ($\delta_{nucleus}-{Փ}_{drx}$) over time. $\delta_{nucleus}$ is the initial size of a DRX nucleus and follows (Equation (19)):(19)$$\delta_{nucleus}=\frac{1}{\kappa \sqrt{{\rho_{i,\gamma }+\rho }_{m,\gamma }}+\frac{1}{{Փ}_{0}}}$$
where ${Փ}_{0}$ and κ are the initial grain size of the undeformed microstructure and a constant, respectively (see Table A1 in Appendix B).

#### 3.6.3. Recrystallisation Fraction

The recrystallisation fraction, X, follows the JMAK equation [22], assuming that recrystallisation grains are spherical (Equation (20)).
(20)$$X\left(t\right)=1-\exp\left(-\frac{4\pi }{3}N{Փ}_{drx}^{3}\right)$$
where N is the number of nuclei at each strain step.

Once DRX initiates, new dislocation-free grains nucleate, and the dislocation density and the flow stress change. Therefore, we introduce an average dislocation density $\rho_{ave}$ representing the total dislocation density in austenite considering the recrystallisation and the non-recrystallised portions of material (Equation (21)):(21)$$\rho_{ave}=\left(1-X\right)\left(\rho_{i,\gamma }+\rho_{m,\gamma }\right)+X.\rho_{ss,drx}$$
where $\rho_{ss,drx}$ is the dislocation density at the steady state in a fully recrystallised microstructure, given in Appendix B, Table A2.

The sum of the dislocation density in ferrite and austenite is the total dislocation density of the steel and reads (Equation (22)):(22)$$\rho_{total}=f_{\alpha }\left(\rho_{i,\alpha }+\rho_{m,\alpha }\right)+\left(1-f_{\alpha }\right)\rho_{ave}\;\;\;$$

#### 3.6.4. Average Grain Size Model

The size of a recrystallised grain that nucleated at the time $t_{1}$ changes over time. The grain size at $t_{2}>t_{1}$ can be calculated by integrating the growth rate of the recrystallised grain, ${\dot{Փ}}_{drx}=\frac{{dՓ}_{drx}}{dt}$, in the time interval of ($t_{2}$-$t_{1}$) follows (Equation (23)):(23)$${Փ}_{drx}=\int_{t_{1}}^{t_{2}}{\dot{Փ}}_{drx}dt\;$$

The average grain size, ${Փ}_{ave}$, taking into account both the population of non-recrystallised and recrystallised grains, reads (Equation (24)):(24)$${Փ}_{ave}={Փ}_{0}\left[1-X\left(t\right)\right]+{Փ}_{drx}\cdot X\left(t\right)\;$$

### 3.7. Load Partitioning between Ferrite and Austenite

The strain rate at which each phase deforms is determined based on the iso-work approach proposed by Bouaziz [24]. This approach assumes that the amount of mechanical work increment in each phase remains constant at each strain step. The iso-work model considers simple mixture law for strain rate as follows (Equation (25)):(25)$$\dot{\varepsilon }=f_{\alpha }{\dot{\varepsilon }}_{\alpha }+\left(1-f_{\alpha }\right){\dot{\varepsilon }}_{\gamma }\;$$
where subscripts α and γ correspond to ferrite and austenite, respectively, $f_{\alpha }$ is ferrite volume fraction, and $\dot{\varepsilon }$ is the global strain rate. Equal mechanical work in both phases gives (Equation (26)):(26)$$\sigma_{\alpha }{\dot{\varepsilon }}_{\alpha }=\sigma_{\gamma }{\dot{\varepsilon }}_{\gamma }\;$$

Replacing strain rate in ferrite with ($\sigma_{\gamma }{\dot{\varepsilon }}_{\gamma }/\sigma_{\alpha }$) and reorganising (25), the strain rate in austenite reads (Equation (27)
(27)$${\dot{\varepsilon }}_{\gamma }=\frac{\dot{\varepsilon }}{f_{\alpha }\left(\frac{\sigma_{\gamma }}{\sigma_{\alpha }}\right)+1-f_{\alpha }}$$

Similarly, the strain rate in the ferrite phase follows (Equation (28)):(28)$${\dot{\varepsilon }}_{\alpha }=\frac{\dot{\varepsilon }\left(\frac{\sigma_{\gamma }}{\sigma_{\alpha }}\right)}{f_{\alpha }\left(\frac{\sigma_{\gamma }}{\sigma_{\alpha }}\right)+1-f_{\alpha }}$$

Finally, the model computes the overall flow stress, σ, for the evolving phase volume fraction and the respective stress of ferrite and austenite through a mixture rule as follows (Equation (29):(29)$$\sigma =f_{\alpha }\sigma_{\alpha }+\left(1-f_{\alpha }\right)\sigma_{\gamma }\;$$

## 4. Results

### 4.1. Compression Flow Curves

Figure 4 shows the influence of the strain rate and temperature on the flow stress evolution over the strain. During deformation at low temperatures, the flow stress increases and reaches a steady state. This behaviour represents work hardening and dynamic recovery in ferrite during hot deformation. Moreover, the temperature is not high enough to provide the driving force for DRX initiation. We observed an unusual behavior at 650 °C and 700 °C at the lowest strain rate (respective solid curve, Figure 4c). During deformation, the flow stress decreases after reaching a peak value and then saturates at lower stresses. This behaviour was also observed at 650 °C and 0.01 $s^{-1}$. At this particular deformation condition, the peak strain decreases with decreasing the deformation temperature, as shown in Figure 5. This softening at low temperatures is attributed to ferrite formation during deformation. On the other hand, deforming at slow strain rates enhances both DRV and ferrite formation due to the longer deformation times. Therefore, the presence of ferrite reduces the overall flow stress [25]. Wray [26] reported a similar behaviour in ferrite and austenite flow stress as temperature decreased. Figure 6a illustrates the evolution of strain rate in phases deformed at 750 °C at the strain rate of 0.01 $s^{-1}$. The strain rate decreases in ferrite at the beginning of deformation (strain < 0.05), implying low work hardening (Equation (13)), as shown by a decreased hardening parameter in this phase in Figure 6b. As deformation proceeds, the volume fraction of ferrite increases, raising the strain rate in this phase (Equation (28)), and thereby promoting work hardening of ferrite. The work hardening parameter in ferrite at the strain rate of 0.01 $s^{-1}$ (black dotted curve) is higher than that at 0.001 $s^{-1}$ (black solid curve). Moreover, the softening parameter in ferrite at the strain rate of 0.001 $s^{-1}$ (red solid curve) is higher than that at 0.01 $s^{-1}$ (red dotted curve). These observations suggest promoted DRV under slow deformation. A higher strain rate promotes work hardening, raising the strength of ferrite closer to that of austenite, as shown in Figure 6c.

At temperatures above 800 °C, the flow stress curves exhibit a single peak stress followed by a strain-softening before reaching a steady state at the end of the deformation. The flow curves’ characteristics recall the DRX phenomenon during hot deformation. The peaks shift to smaller strains as the deformation temperature increases because elevated temperatures provide a higher driving force for the nucleation and growth steps. Furthermore, the peak stress and the peak strain decrease with decreasing strain rate at higher temperatures (see Figure 5). The flow curves below 850 °C show a peak characteristic due to softening caused by ferrite formation.

Figure 7 shows the relationship between peak strain and processing conditions above 800 °C through the Zener-Hollomon parameter.

### 4.2. Microstructure of Deformed Samples

Figure 8 shows the microstructural modification of the studied microalloyed steel after compression at various temperatures and at a strain rate of 0.01 $s^{-1}$ up to a strain of 0.8. At 650 °C, the microstructure consists primarily of ferrite with a low amount of martensite formed after quenching the austenite phase (indicated by black arrows in Figure 8a). Increasing the deformation temperature, the microstructure presents elongated austenite grains and a ferrite phase formed during cooling at the prior austenite grain boundaries (indicated by black arrows in Figure 8b,c). At 800 °C, austenite grain boundaries become serrated due to stored energy of deformation and high variation in dislocation density at these regions. As the temperature rises, the mobility of the high-angle grain boundaries increases, triggering the nucleation of recrystallised grains at the prior austenite grain boundaries (indicated by blue arrows in Figure 8d and corresponding higher magnification image Figure 8g). At higher temperatures, DRX proceeds, and the microstructure after deformation at 850 °C consists of austenite recrystallised grain, $\gamma_{drx}$, (indicated by red arrows in Figure 8e and corresponding higher magnification image Figure 8h). The recrystallised grain size increases with increasing deformation temperature because the elevated temperatures enhance growth over nucleation rate (Figure 8f and corresponding higher magnification image Figure 8i).

Figure 9 displays the microstructure of the samples deformed at 700 °C under two strain rates up to a strain of 0.8. The microstructures consist of ferrite (indicated by black arrows in Figure 9) and martensite, M, transformed from austenite during quenching (indicated by red arrows in Figure 9). As mentioned in Section 4.1, ferrite amounts increase with slower deformation. The micrographs reveal that at the lowest strain rate, 0.001 $s^{-1}$, the substructure in ferrite becomes more well-defined compared to the specimen deformed at a higher strain rate. Poletti et al. showed the substructure forming in ferrite during hot deformation using EBSD [27]. This observation suggests that increasing strain rate hinders DRV in ferrite, thereby increasing the ferrite strength and reducing strain concentration in these soft regions. 

Figure 10 shows the influence of the strain rate on the microstructure of specimens deformed at 850 °C to strain 0.5 at different strain rates. During deformation, grain boundaries become serrated (indicated by black arrows in Figure 10a,c) due to the pressure opposed on the boundaries. High local stored energy close to grain boundaries is the driving force for the nucleation of new grains. The recrystallised grains, hence, form due to the movement of the bulged boundaries (Figure 10a,c, indicated by red arrows). As the strain rate decreases, the stored energy decreases due to lower dislocation density. However, this reduction in strain rate provides more time for grain boundary migration and lower critical dislocation density for DRX initiation [20], promoting DRX at a given temperature (Figure 10b,d). Under both strain rates, a small number of recrystallised grains were observed (indicated by blue arrows in Figure 10c,d).

Figure 11 shows the microstructure modifications for austenite deformed at 900 °C and 0.01 $s^{-1}$ up to various strains of 0.2, 0.4, 0.5, and 0.8. Before deformation, i.e., at strain 0, the initial microstructure consists of equiaxed coarse austenite grains, as illustrated in Figure 3b. In the early stages of deformation, dislocation density increases near the grain boundaries, causing the grain boundary to bulge (indicated by red arrows in Figure 11a). The grain boundaries migrate and become serrated due to applied strain, known as the strain-induced grain boundary migration mechanism, SIBM. Figure 11b,c show bulged grain boundaries and some recrystallised grains. With further straining, the fraction of DRX grain increases until the microstructure gets almost fully recrystallised at strain 0.8. The original distribution of grains is completely modified, and the microstructure consists of new equiaxed recrystallised grains; see Figure 11d.

### 4.3. Dynamic Recrystallisation Fraction

Figure 12 depicts the modelled evolution of the DRX fraction. At a given strain rate, the fraction of recrystallised grain increases as temperature increases. The initiation of the DRX shifts to smaller strains at higher temperatures and lower strain rates; see Figure 12a–c. High temperatures and slower deformation enhance recrystallisation. At a given deformation temperature, the fraction of recrystallisation increases by decreasing the deformation rate; see Figure 12d.

### 4.4. Average Grain Size

Figure 13 depicts the modelled evolution of the average grain size using Equation (24). The average grain size considers the population of recrystallised and unrecrystallised grain. For a given strain rate and temperature, the average grain size decreases as deformation proceeds because the fraction of DRX increases. The average grain size increases as the temperature increases due to the higher population of coarser DRX grains, contributing to the average grain size. Figure 13d illustrates the evolution of the average grain size within samples deformed at 900 °C and different strain rates. At the highest strain rates, the time is insufficient to initiate DRX, the fraction of recrystallised grains is trivial, and the average grain size shows no significant change over the strain. During slow deformation, the prolonged time allows for deformation energy accumulation and grain boundary movement, thereby enhancing DRX. The average grain size reaches a higher steady-state value at a deformation rate of 0.001 s^−1^ compared to 0.01 s^−1^. This is attributed to the reduced availability of nucleation sites at the lowest strain rate due to low stored energy. In addition to that, the grain boundary migration and subsequent grain growth are more advanced at the same strain when the strain rate is slow.

Figure 14 compares the modelled and experimental average grain size. The model successfully describes the tendency of the mean grain size with the strain rate and temperature. It is important to note that if the recrystallised and the deformed grains cannot be distinguished due to their similar sizes, the resulting quantification is less accurate.

### 4.5. Phase Volume Fraction

Figure 15a and Figure 15d show the microstructure of the samples after heat treatments at 750 °C for 80 s and 800 s. These holding times represent the duration of deformation at 750 °C for samples reaching a final strain of 0.8 at strain rates of 0.01 $s^{-1}$ and 0.001 $s^{-1}$. We measured the ferrite volume fraction using OM images and ImageJ 1.53e software. The ferrite volume fraction is 0.29 ± 0.01 in the sample heat-treated at 750 °C for 800 s, 0.37 ± 0.01 in the sample deformed to a strain of 0.8 at 750 °C and 0.001 $s^{-1}$, and 0.31 in the sample cooled to 750 °C at a rate of 1 °C/s as simulated using JMatPro v14 software. Therefore, we concluded that the difference in ferrite volume fraction between the heat-treated sample held at the deformation temperature for an equivalent time and the deformed microstructure represents the ferrite formed during isothermal deformation. Figure 15 displays the evolution of ferrite with increasing strain for the specimens deformed at 750 °C under strain rates of 0.01 $s^{-1}$ and 0.001 $s^{-1}$. The micrographs show that the amount of ferrite increases with strain. Furthermore, comparing two strain rates, ferrite decreases with faster deformation due to less time for transformation.

Figure 16 presents a comparison between the ferrite volume fraction of the specimen deformed at 750 °C under strain rates of 0.001 $s^{-1}$ and 0.01 $s^{-1}$ at the end of deformation, i.e., strain of 0.8, obtained by the model and experiments. The undeformed condition refers to the sample quenched before deformation, as described in Section 2.3.

Figure 17 shows the modelled evolution of the ferrite volume fraction at various deformation conditions using Equation (4) and the experimental results using OM images and ImageJ 1.53e software. Ferrite volume fraction increases with decreasing temperature and strain rate because the ferrite nucleation is a diffusion-controlled transformation. Lower deformation temperatures provide a higher driving force for phase transformation, whereas slower strain rates provide more time for this diffusional phenomenon.

### 4.6. Dislocation Density Evolution

Figure 18 illustrates the modelled evolution of total dislocation density, comprising the sum of ferrite and austenite dislocation densities. At the beginning of the deformation, the dislocations multiplicate, causing work hardening. Further straining at lower temperatures decreases the work hardening rate due to the dislocation annihilation through DRV. Once the rate of work hardening and DRV counterbalance, the dislocation density reaches equilibrium and remains constant. On the other hand, at higher temperatures, after reaching the critical strain, the softening proceeds and the dislocation density decreases due to the DRX process. The dislocation density reaches lower values at higher temperatures. Furthermore, the total dislocation density has higher values at higher strain rates due to less DRX.

Figure 18d compares the evolution of the total dislocation density of deformation at 1000 °C under various strain rates. The total dislocation density is lower at lower strain rates due to progressive DRX at this temperature.

## 5. Discussion

In this section, we correlate the information obtained from compression tests with the modification in the microstructure responsible for damage during hot tensile tests.

### 5.1. Microstructure Evolution

In the two-phase domain, the microstructure consists of ferrite and austenite. As the temperature increases within this domain, the amount of ferrite decreases, becoming confined to the boundaries of elongated austenite grains. Elevated temperatures enhance the kinetics of DRX. At high temperatures, bulging features appear at austenite grain boundaries, which act as nucleation sites for DRX. Figure 19 compares the microstructure of specimens compressed at 900 °C and 1100 °C at a strain rate of 0.01 $s^{-1}$ up to a strain of 0.2. The specimen deformed at 900 °C shows early stages of DRX (indicated by black arrows showing bulged boundaries in Figure 19a) with minimal progression (less than 0.05%), consistent with the modelled results in Figure 12b, whereas at 1100 °C, the specimen is fully recrystallised at a strain of 0.2 (Figure 19b). Furthermore, the deformation speed impacts the amount of ferrite and the associated restoration mechanisms, such as DRV in ferrite and primarily DRX in austenite. Lower strain rate promotes dislocation annihilation by DRV in ferrite (Equation (14) and Figure 6), resulting in softer ferrite with a well-defined substructure (Figure 9). Lower strain rates prolong the time for energy accumulation, which serves as a driving force for recrystallisation in austenite, promoting DRX in this phase (Figure 10). In addition, lower strain rates and elevated temperatures decrease the critical strain required for DRX, implying that DRX initiates at smaller strains under such conditions as shown in Figure 7.

### 5.2. Effect of the Microstructure Evolution on the Ductility Minimum

In this section, we correlate the deformation behaviour of the studied alloy with the ductility under tension. In previous works on this material [6,28,29], we observed that: Hot ductility is high in the austenite range. It decreases close to ferrite formation and increases again when the ferrite amount increases.Hot ductility improves with increasing the strain rate.The crack starts at the prior austenite grains, with or without MnS particles or ferrite.

We have to consider three main aspects of the microstructure modification that have an impact on the damage during tensile deformation:


**Discontinuous dynamic recrystallisation**


At elevated temperatures, DRX changes the microstructure by releasing stored energy from dislocations through the movement of high-angle grain boundaries. During hot deformation, DRX lowers dislocation density and deformation energy, resulting in decreased flow stress and strain hardening. This softening effect reduces stress concentration and improves ductility.


**Grain boundary sliding**


Grain boundary sliding is promoted at longer deformation times, meaning slower deformation rates. Grain boundary sliding is the dominant fracture mechanism for steels at high temperatures [7,30,31]. Figure 20 shows the OM images of hot tensile specimens deformed in the austenitic range at 850 °C under two strain rates of 0.01 $s^{-1}$ and 0.001 $s^{-1}$. The micrographs of the fractured samples show that length and the number of intergranular damage increase during slow deformation, suggesting that an increasing strain rate improves the hot ductility. Similar findings were observed for the same material in [29] for the samples strained at 850 °C under two strain rates of 0.01 $s^{-1}$ and 0.001 $s^{-1}$ until rupture.


**Ferrite formation**


Figure 21 illustrates the modelled strain in the ferrite phase when the samples are deformed in the two-phase field. In this domain, the amount of ferrite increases as the temperature and strain rate decrease. Lower temperatures enhance ferrite formation due to higher undercooling, which provides a driving force for ferrite nucleation. Additionally, lower strain rates in the two-phase domain allow more time for ferrite nucleation, further increasing the amount of ferrite. Furthermore, higher strain rates enhance work hardening in ferrite, increasing the strength of this phase (see Figure 6). As deformation proceeds, the volume fraction of ferrite increases. A higher ferrite fraction distributes the strain more uniformly within this phase, allowing the sample to deform further before fracture [12]. At lower strain rates, the strength of ferrite reduces due to promoted dislocation annihilation (DRV). As a result, strain concentrates in this softer phase, leading to fracture. Figure 22 shows the microstructure of tensile specimens strained at 750 °C under two strain rates up to rupture. At low strain rates, damage initiates within the ferrite at austenite grain boundaries (see Figure 22b) due to enhanced DRV and softening of ferrite. In contrast, under rapid deformation, damage initiates at the interface between austenite and ferrite (indicated by red arrows in Figure 22a), indicating less softening and strain concentration in ferrite.

## 6. Conclusions

We investigated the hot deformation behaviour of microalloyed steels through compression tests. Understanding the hot deformation in the absence of necking, we aimed to connect how the microstructural changes affect the ductility under tensile deformation as in continuous casting processes. We conclude that:In the two-phase domain, increasing the strain rate enhances ductility by reducing the fraction of the ferrite formed and promoting work hardening of ferrite by retarding the annihilation of dislocations by DRV.In addition to ferrite volume fraction, the ductility improves at higher strain rates in the two-phase field because there is less time for grain boundary sliding.Above the transformation temperature, the ductility increases due to the occurrence of DRX as the temperature increases.DRX modifies the microstructure during hot deformation by reducing dislocation density and deformation energy. This process decreases flow stress and strain hardening, reducing stress concentrations and improving ductility at elevated temperatures.In the austenitic range, at higher strain rates, ductility tends to increase due to reduced grain boundary sliding even without DRX. However, excessive grain boundary sliding at low strain rates makes DRX less effective in influencing ductility.

## Figures and Tables

**Figure 2 materials-17-04551-f002:**
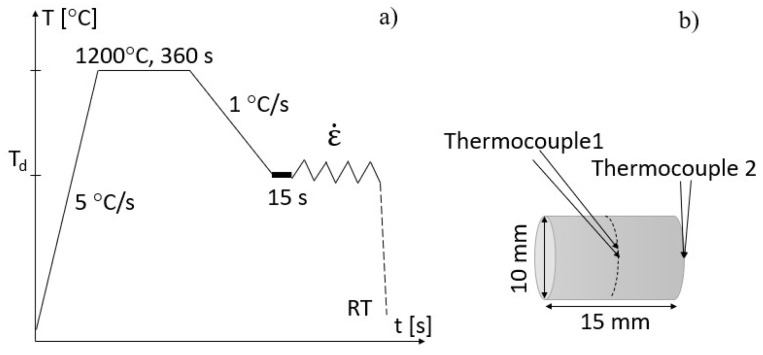
(**a**) Thermomechanical route of hot compression, (**b**) Schematic of the compression test specimen.

**Figure 3 materials-17-04551-f003:**
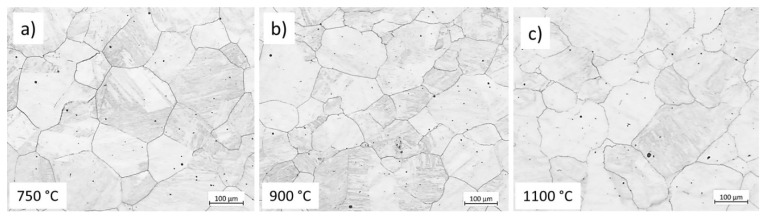
Initial microstructure (before deformation) of the samples held for 6 min at 1200 °C, cooled down to (**a**) 750 °C, (**b**) 900 °C, (**c**) 1100 °C, and quenched with water and etched with CRIDA. The scale bars represent 100 µm.

**Figure 4 materials-17-04551-f004:**
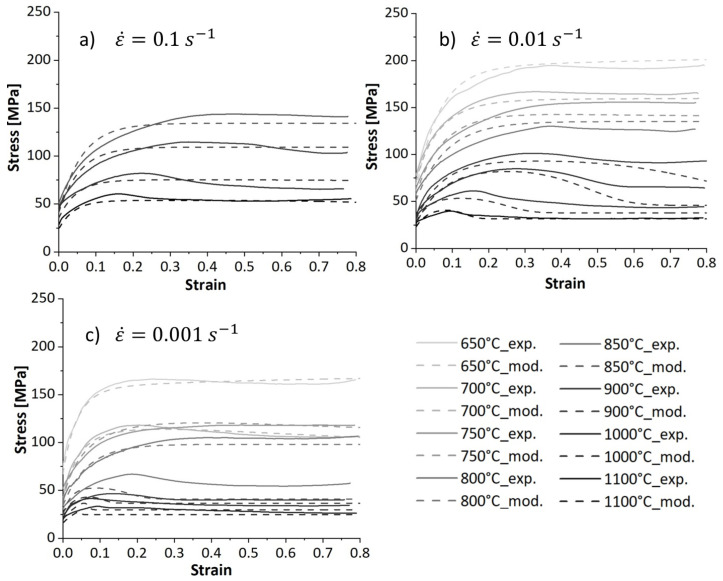
True stress-true strain curves of studied microalloyed steel deformed at different temperatures and strain rates of (**a**) 0.1 s^−1^, (**b**) 0.01 $s^{-1}$, (**c**) 0.001 $s^{-1}$.

**Figure 5 materials-17-04551-f005:**
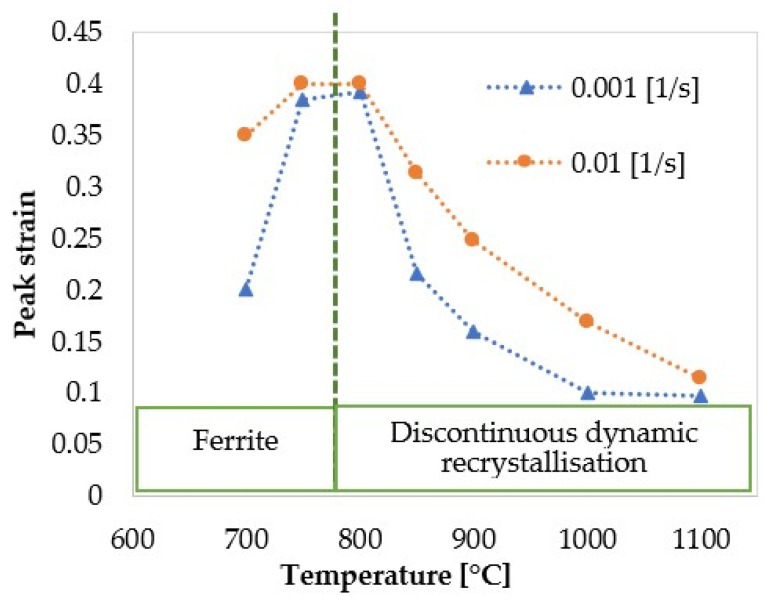
The relation between peak strain and temperatures at two tested strain rates.

**Figure 6 materials-17-04551-f006:**
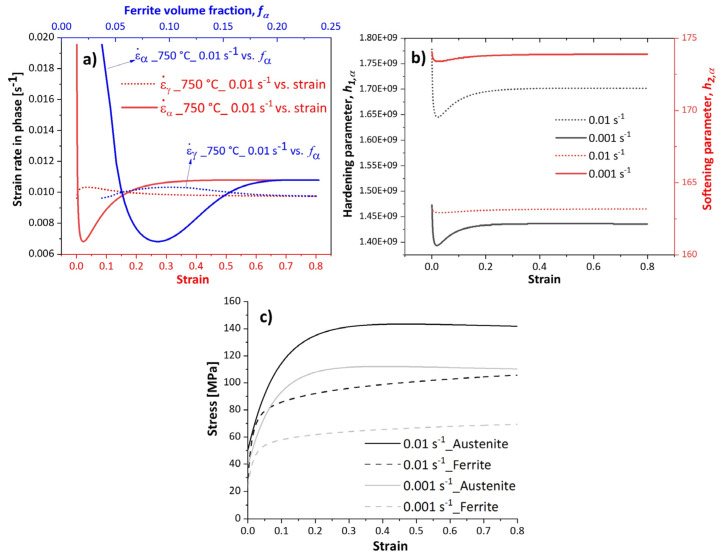
(**a**) Evolution of strain rate, (**b**) evolution of hardening and softening parameters, and (**c**) Modelled flow curves of the ferrite and austenite of the samples deformed at 750 °C and strain rates of 0.01 $s^{-1}$ and 0.001 $s^{-1}$.

**Figure 7 materials-17-04551-f007:**
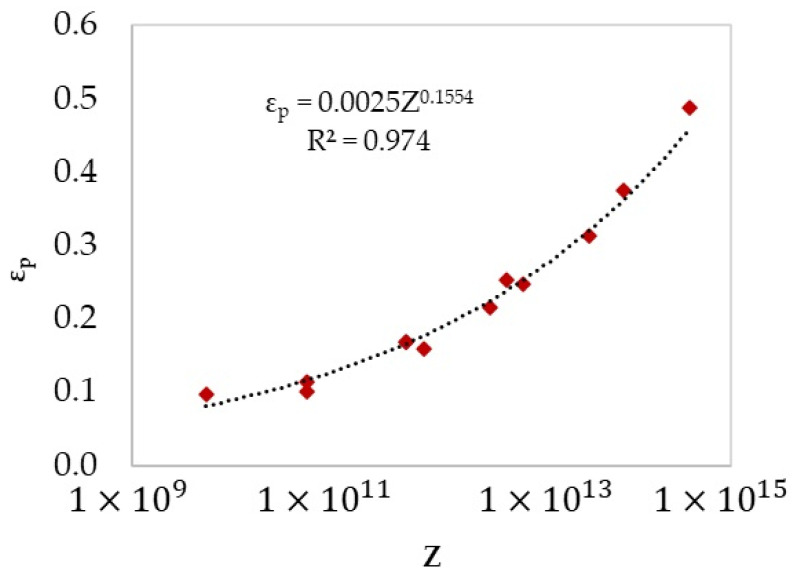
Plot of peak strain versus Z parameter from the flow curves above 800 °C.

**Figure 8 materials-17-04551-f008:**
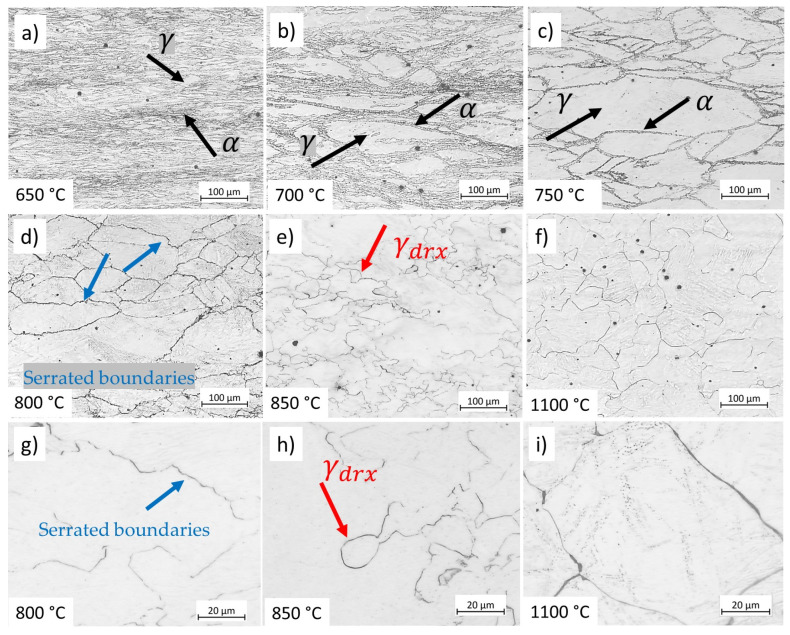
Influence of temperature on the microstructure of deformed microalloyed steel at a strain rate of 0.01 $s^{-1}$ at strain 0.8 and: (**a**) 650 °C, (**b**) 700 °C, (**c**) 750 °C, (**d**) 800 °C, (**e**) 850 °C, (**f**) 1100 °C. (**g**–**i**) higher magnification images corresponding to (**d**–**f**) respectively. Etchant: CRIDA. The scale bars represent 100 µm.

**Figure 9 materials-17-04551-f009:**
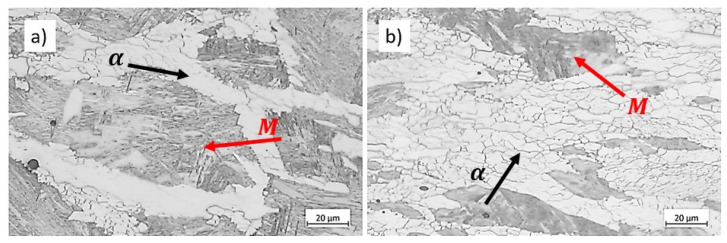
Microstructure of specimens compressed at 700 °C up to strain 0.8 at strain rates of (**a**) 0.01 $s^{-1}$, and (**b**) 0.001 $s^{-1}$. Etchant: Nital. The scale bars represent 20 µm.

**Figure 10 materials-17-04551-f010:**
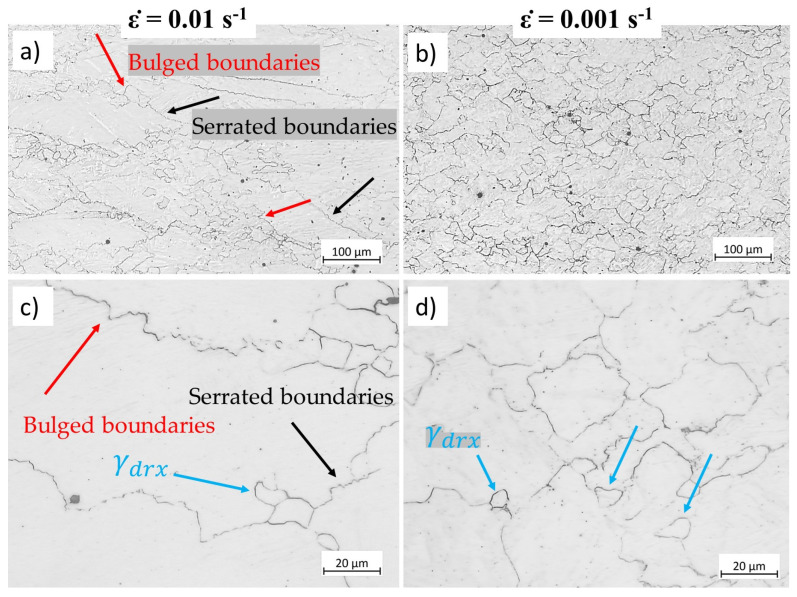
Microstructure of specimens deformed at 850 °C up to strain 0.5 at strain rates of (**a**) 0.01 $s^{-1}$, and (**b**) 0.001 $s^{-1}$. (**c**,**d**) higher magnification images corresponding to (**a**,**b**), respectively. Etchant: CRIDA. The scale bars represent 100 µm.

**Figure 11 materials-17-04551-f011:**
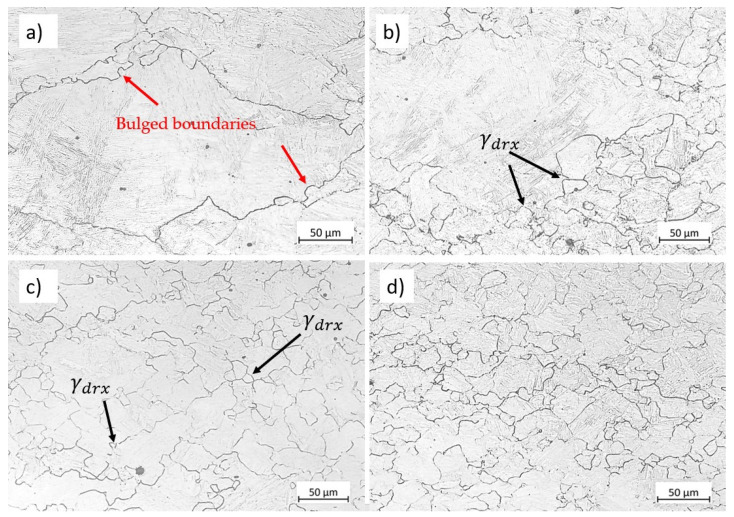
Micrographs of the specimens deformed at 900 °C and 0.01 $s^{-1}\;$ up to various strains: (**a**) 0.2, (**b**) 0.4, (**c**) 0.5, and (**d**) 0.8. Etchant: CRIDA. The scale bars represent 50 µm.

**Figure 12 materials-17-04551-f012:**
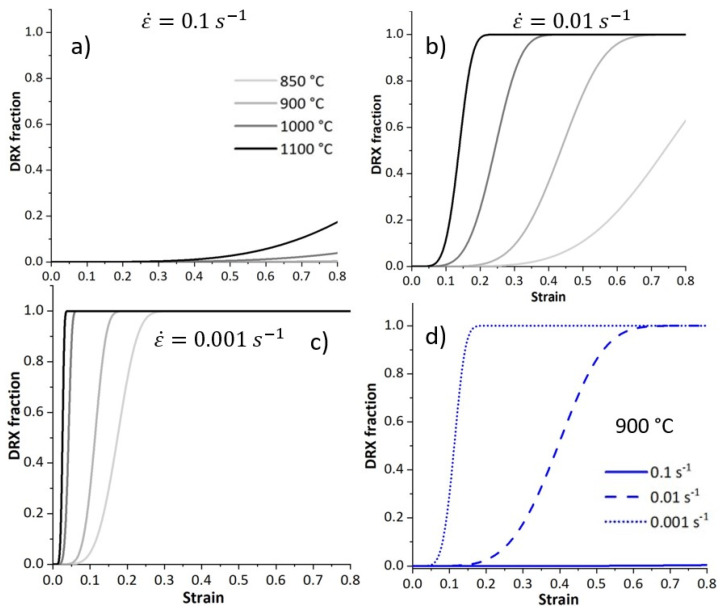
Modelled results representing the evolution of DRX fraction within the specimens deformed at different temperatures and strain rates of (**a**) 0.1 s^−1^, (**b**) 0.01 s^−1^, (**c**) 0.001 s^−1^, and (**d**) modelled DRX fraction of the specimen deformed at 900 °C at various strain rates.

**Figure 13 materials-17-04551-f013:**
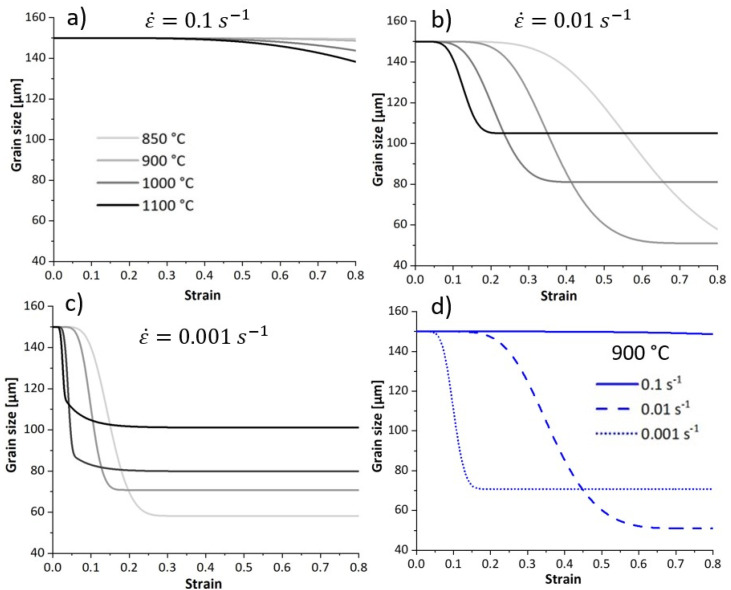
Modelled results representing the evolution of average grain size of specimens deformed at different temperatures and strain rates of (**a**) 0.1 s^−1^, (**b**) 0.01 s^−1^, (**c**) 0.001 s^−1^, and (**d**) modelled average grain size of the specimen deformed at 900 °C and various strain rates.

**Figure 14 materials-17-04551-f014:**
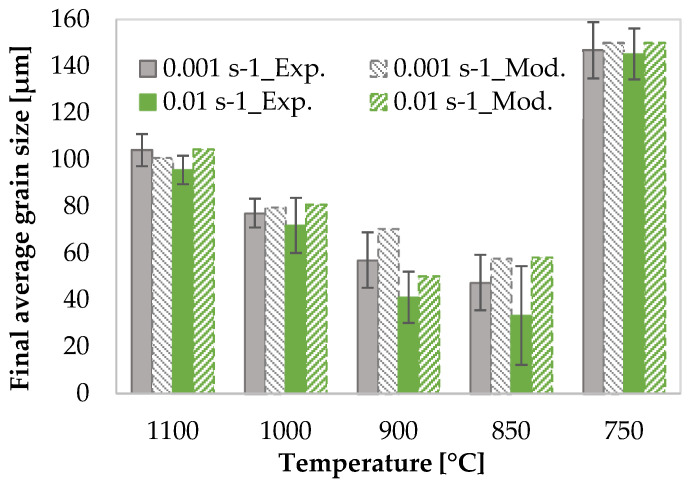
Comparison of modelled and experimental final average grain size for specimens deformed to a final strain of 0.8 at various deformation conditions.

**Figure 15 materials-17-04551-f015:**
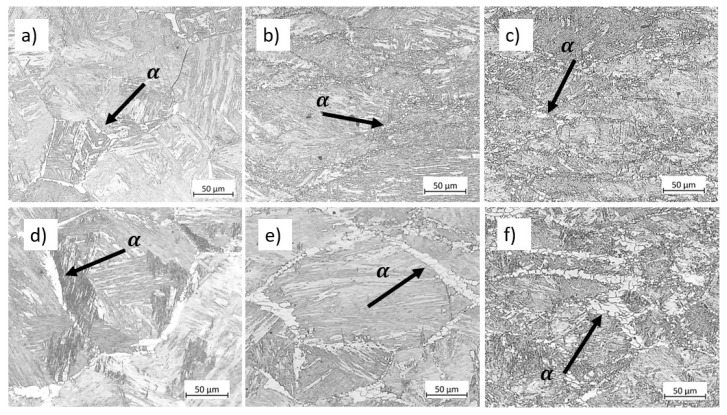
Ferrite within the specimens deformed at 750 °C to a strain of (**a**) 0 held for 80 s, (**b**) 0.5, (**c**) 0.8, (**d**) 0 held for 800 s, (**e**) 0.5, and (**f**) 0.8. Etchant: Nital 3%. The scale bars represent 50 µm.

**Figure 16 materials-17-04551-f016:**
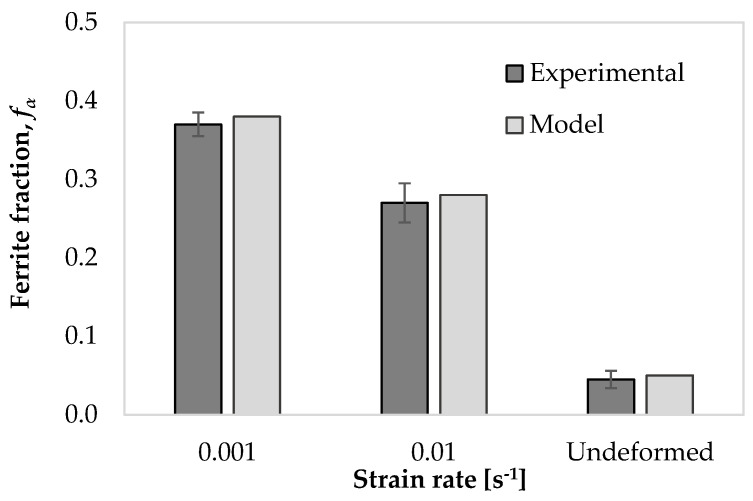
Comparison between different methods for calculating ferrite volume fraction within the undeformed sample and samples deformed at 750 °C under two strain rates at ε = 0.8.

**Figure 17 materials-17-04551-f017:**
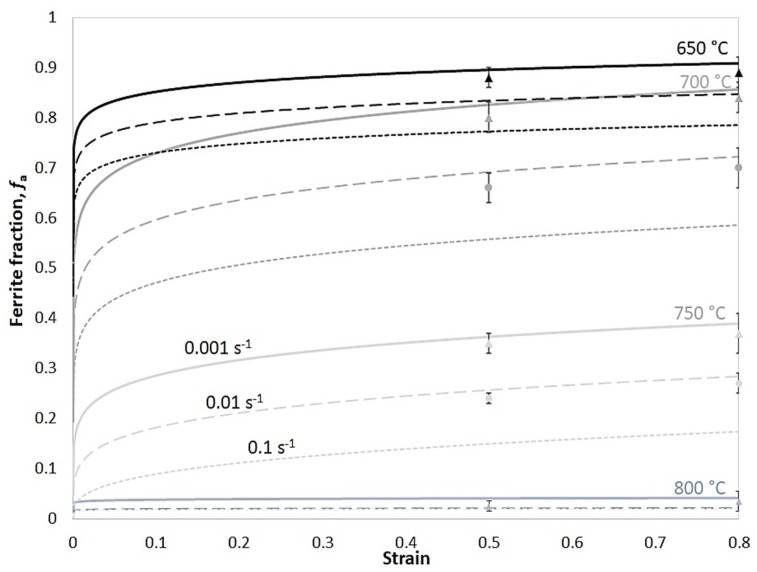
The modelled (continuous curves) and experimental (points) evolution of the ferrite fraction of the specimens deformed at different temperatures and strain rates.

**Figure 18 materials-17-04551-f018:**
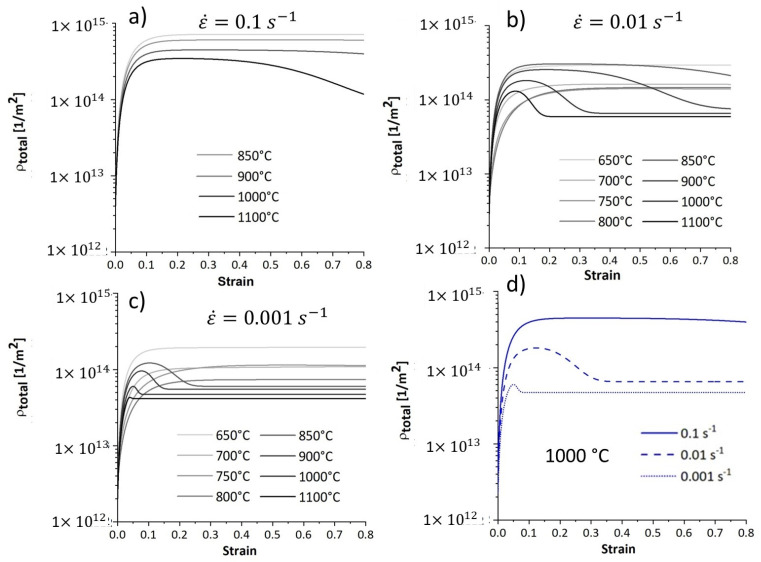
(**a**–**d**) Modelled evolution of total dislocation density of specimens deformed at different temperatures and strain rates.

**Figure 19 materials-17-04551-f019:**
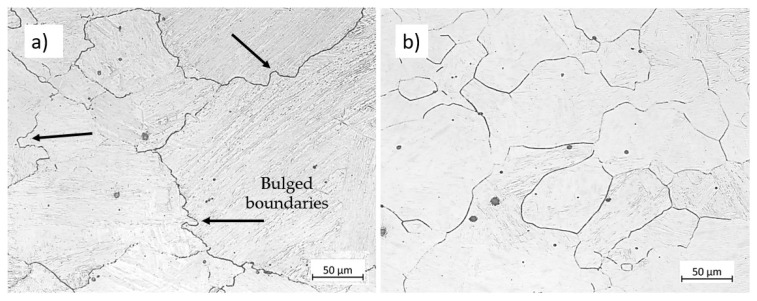
Micrographs of the specimens compressed at the strain rate of 0.01 $s^{-1}$ to the strain of 0.2 at (**a**) 900 °C and (**b**) 1100 °C. Etchant: CRIDA. The scale bars represent 50 µm.

**Figure 20 materials-17-04551-f020:**
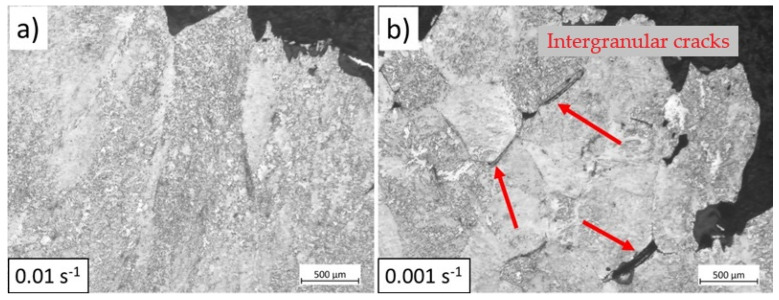
OM images of hot tensile samples deformed at 850 °C under a strain rate of (**a**) 0.01 $s^{-1}$ and (**b**) 0.001 $s^{-1}$. The scale bars represent 500 µm.

**Figure 21 materials-17-04551-f021:**
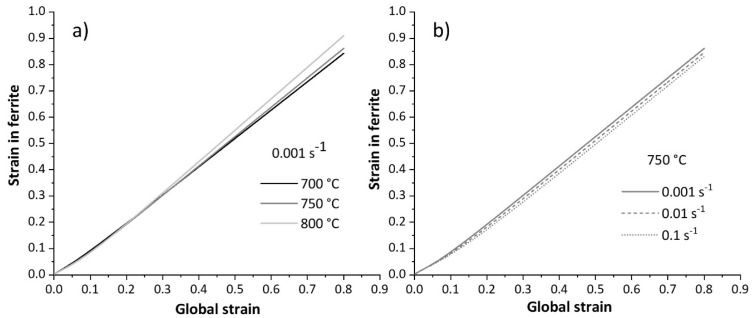
Modelled strain evolution in ferrite within the samples deformed in two-phase domain at different (**a**) temperatures and (**b**) strain rates.

**Figure 22 materials-17-04551-f022:**
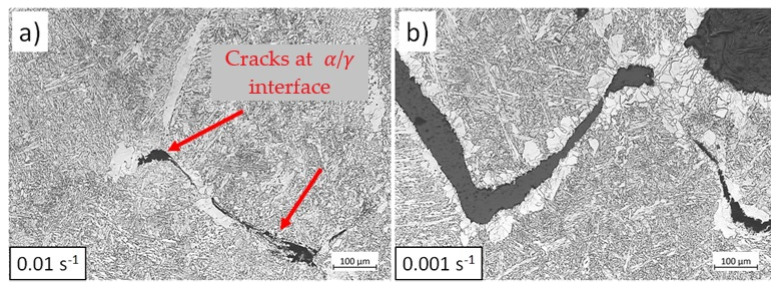
OM images of hot tensile samples deformed at 750 °C under a strain rate of (**a**) 0.01 $s^{-1}$ and (**b**) 0.001 $s^{-1}$. Etchant: Nital. The scale bars represent 100 µm.

**Table 1 materials-17-04551-t001:** Chemical composition of the studied microalloyed steel [wt%].

C	Si	Mn	S	P	Ni	Cr	Al	N	Nb	Ti	B	Fe
0.08	0.12	1.7	0.005	0.01	0.03	0.3	0.05	0.005	0.002	0.001	0.0002	bal.

**Table 2 materials-17-04551-t002:** Experiment metrics for hot compression and phase transformation models.

Temperature [°C]	Strain Rate [s^−1^]	Final Strain [-]
650	0.001	0.5, 0.8
	0.01	0.5, 0.8
700	0.001	0.5, 0.8
	0.01	0.5, 0.8
750	0.001	0.5, 0.8
	0.01	0.5, 0.8
800	0.001	0.5, 0.8
	0.01	0.5, 0.8
850	0.001	0.5, 0.8
	0.01	0.5, 0.8
	0.1	0.5, 0.8
900	0.001	0.5, 0.8
	0.01	0.1, 0.2, 0.4, 0.5, 0.8
	0.1	0.5, 0.8
1000	0.001	0.5, 0.8
	0.01	0.5, 0.8
	0.1	0.5, 0.8
1100	0.001	0.5, 0.8
	0.01	0.1, 0.2, 0.5, 0.8
	0.1	0.1, 0.2, 0.5, 0.8

**Table 3 materials-17-04551-t003:** Phase transformation model fitting parameters.

$m_{a}$	$n_{a}$	ξ	$m_{b}$	$n_{b}$	ψ
−6.18	−0.084	$1.43\times 10^{17}\;\left[\frac{s}{{}^{\circ}C}\right]$	18.26	0.097	$1.30\times 10^{-53}\;\left[\frac{s}{{}^{\circ}C}\right]$

## Data Availability

The original contributions presented in the study are included in the article/Appendix A, further inquiries can be directed to the corresponding author/s.

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
