# Peer review of "The Influence of Thermomechanical Conditions on the Hot Ductility of Continuously Cast Microalloyed Steels"

_materials, 2024, doi:10.3390/ma17184551_

Round 1
Reviewer 1 Report
Comments and Suggestions for Authors
The manuscript entitled The influence of Thermomechanical Conditions on the hot ductility of steels presented to be published in Materials is well-written and the main goal and results presented along the manuscript agree with the aim of the Journal. However, previously to be published in this journal the following questions and comments may be solved:
· In the abstract, the authors define the acronyms dDRX and DRV; however, only one time is used; in this case it is necessary to remove the acronyms from the abstract section. Also, they are talking about DRX, however this acronym is not defined.
· In the introduction section, I miss a contextualisation of the manuscript. Which is the main difference between all the work presented in the literature and the once you are trying to publish in this journal. A paragraph just after Figure 1 may be included in order to clarify this point
· In the material section, the material you use is commercial, if yes include the required information
· Take care with the equations, see equations 21, 22, 27 and 28 (equation may be located in one line)
· Figure 4c, the y-axis may be changed in agreement with the y-value for figure 4a and 4b (250 MPa)
· Figure 5 shows the peak strain (I understand the y-axis does not present any units) vs. temperature). The line between the points may be dash, you did not conduct any experiment between the represented temperatures
· The authors in the microstructure section, talk about serrated boundaries and also martensite/austenite phase; please provide a high magnification SEM micrograph in order to clarify this point. Figures 8, 9 and 10 are not enough and the quality of the micrographs does not lead to better observation of this findings and following the discussion of the manuscript
· Why the authors do not provide a phase EBSD map in order to prove the dynamic recrystallisation fraction as well as the phase volume fraction; this observation will lead to perfectly close the research conducted in this manuscript
· Furthermore, is clear that the dislocation density is strain rate dependent, however it will be good if the data presented in section 4.6 are linked with the microstructural changes. Also, I consider that if the authors can include a TEM or EBSD micrograph proving the density evolution will lead to increase the quality of this manuscript
Comments on the Quality of English LanguageThe english present along the manuscript is appropiate, several minor changes may be done. Please review the manuscript prior to be re-submitted
Author Response
The manuscript entitled The influence of Thermomechanical Conditions on the hot ductility of steels presented to be published in Materials is well-written and the main goal and results presented along the manuscript agree with the aim of the Journal. However, previously to be published in this journal the following questions and comments may be solved:
- In the abstract, the authors define the acronyms dDRX and DRV; however, only one time is used; in this case it is necessary to remove the acronyms from the abstract section. Also, they are talking about DRX, however this acronym is not defined.
The acronyms in abstract are removed.
The acronyms are defined within the manuscript in Section 3.3.
DRX has been defined in Section 3.3 (line 202).
- In the introduction section, I miss a contextualisation of the manuscript. Which is the main difference between all the work presented in the literature and the once you are trying to publish in this journal. A paragraph just after Figure 1 may be included in order to clarify this point.
Thank you for your recommendation. I modified the introduction (lines 60-101) and explained the aim of our research in a better way (above Figure 1).
- In the material section, the material you use is commercial, if yes include the required information.
The material is a continuously cast microalloyed steel with an internal designation but no commercial name. In previous studies has been reported by steel A (https://www.mdpi.com/2075-4701/12/11/1808) and steel number 1 (https://doi.org/10.37904/metal.2021.4140).
- Take care with the equations, see equations 21, 22, 27 and 28 (equation may be located in one line).
Modified.
- Figure 4c, the y-axis may be changed in agreement with the y-value for figure 4a and 4b (250 MPa).
Modified.
- Figure 5 shows the peak strain (I understand the y-axis does not present any units) vs. temperature). The line between the points may be dash, you did not conduct any experiment between the represented temperatures.
Modified to dot and straight lines.
- The authors in the microstructure section, talk about serrated boundaries and also martensite/austenite phase; please provide a high magnification SEM micrograph in order to clarify this point. Figures 8, 9 and 10 are not enough and the quality of the micrographs does not lead to better observation of this findings and following the discussion of the manuscript.
Figure 8: Modified in Section 4.2 (lines 366-372) with higher magnification of the serrated boundaries and recrystallized grains.
Figure 9: This is the highest magnification of LOM. SEM was not used in this work.
Figure 10: modified. Higher magnification images are added.
- Why the authors do not provide a phase EBSD map in order to prove the dynamic recrystallisation fraction as well as the phase volume fraction; this observation will lead to perfectly close the research conducted in this manuscript.
Actually, EBSD analysis was performed. However, the reconstruction technique was not effective due to the challenges in reconstructing parent austenite grains from the martensitic phase. This difficulty arises from the austenite-to-martensite transformation during water quenching. And the Martensite laths brings artifacts in reconstructing the austenite. Therefore, we characterized the microstructure using an etchant that could reveal the austenite grain boundary, and not martensite laths. But I agree. EBSD in austenitic microstructures (without martensitic transformation) for instance stainless steels or Ni superalloys is the most effective way to characterize DRX.
- Furthermore, is clear that the dislocation density is strain rate dependent, however it will be good if the data presented in section 4.6 are linked with the microstructural changes. Also, I consider that if the authors can include a TEM or EBSD micrograph proving the density evolution will lead to increase the quality of this manuscript.
I understand your point and I agree. Dislocation density measurement would be an interesting validation for microstructural modification. EBSD measurement was not effective as I explained in comment number 8. Similarly, TEM, XRD, or synchrotron methods could help, but within the martensitic microstructure it is challenging to acquire a reasonable result.
Comments on the Quality of English Language
The english present along the manuscript is appropiate, several minor changes may be done. Please review the manuscript prior to be re-submitted
Submission Date
22 August 2024
Date of this review
05 Sep 2024 08:27:01
Reviewer 2 Report
Comments and Suggestions for Authors
Dear authors,
Thank you for sending the manuscript, you will find my comments below.
The article discusses the behavior of microalloyed steel at elevated temperatures. The organization of the manuscript is correct, the planned experiments are justified. The title reflects the content of the manuscript. In my opinion, however, the introduction section should be expanded to include other experiments in this area. I recommend avoiding terms such as "We carried out experiments", "We heat-treated", "we conducted", "We performed" etc. in favor of impersonal terms. Please correct Figure 5 - it is inappropriate to use smooth lines in a scientific article. Please consider presenting the effects of simulations of TTT and CCT graphs described in Chapter 3.2.
Author Response
Comments and Suggestions for Authors
Dear authors,
Thank you for sending the manuscript, you will find my comments below.
The article discusses the behavior of microalloyed steel at elevated temperatures. The organization of the manuscript is correct, the planned experiments are justified. The title reflects the content of the manuscript.
- In my opinion, however, the introduction section should be expanded to include other experiments in this area.
I modified the introduction (lines 60-101). I included the work of other group with respect to ferrite films and ductility, and explained the aim of our research in a better way (above Figure 1).
- I recommend avoiding terms such as "We carried out experiments", "We heat-treated", "we conducted", "We performed" etc. in favor of impersonal terms.
Thank you for your feedback. I have aimed to follow the scientific writing standards, inspired by an informative lecture on the topic:
https://www.youtube.com/watch?v=sS-Txm3R3v8
However, I understand the importance of using impersonal language and if required I will make the necessary adjustments.
- Please correct Figure 5 - it is inappropriate to use smooth lines in a scientific article.
Modified to straight lines with markers.
- Please consider presenting the effects of simulations of TTT and CCT graphs described in Chapter 3.2.
I had a few interrupted experiments (with interrupted strains of 0.5 and 0.8) to validate the model (equation 4). I used JMatPro to determine the phase fraction of phases during the respective experiments and making sure that JMatPro can represent the phase fraction in our material with underlying thermomechanical condition. After ensuring this, I used JMatPro for the whole strain range to get the equation 4. Therefore, the model actually shows the CCT and TTT values (figure 16 and 17).
I apologize, but if I understood your comment correctly.
Submission Date
22 August 2024
Date of this review
30 Aug 2024 13:30:39
Reviewer 3 Report
Comments and Suggestions for Authors The paper entitled “The influence of thermomechanical conditions on the hot ductility of steels” presents a relevant study on how microstructural changes decrease the formability of a microalloyed steel during continuous casting in which the authors provide an extensive model and experimental analysis describing the mechanisms of work hardening and dynamic restoration mechanisms through discontinuous dynamic recrystallization in austenite and dynamic recovery in ferrite and austenite. In general, the paper shows a well-described modeled analysis and experimental validation, with some gaps that have been addressed in the review. Some additional observations should be taken into account to improve the paper. 1. The title does not reflect the microalloyed steel material used in the study. Include the specific material in the title of the paper and not just “steels”. 2. Include in the abstract the microalloyed steel material used in the study. 3. The introduction section should be improved by including an extensive analysis of the state of the art available in literature. This section shows a lack of analysis of the state of the art.Author Response
Comments and Suggestions for Authors
The paper entitled “The influence of thermomechanical conditions on the hot ductility of steels” presents a relevant study on how microstructural changes decrease the formability of a microalloyed steel during continuous casting in which the authors provide an extensive model and experimental analysis describing the mechanisms of work hardening and dynamic restoration mechanisms through discontinuous dynamic recrystallization in austenite and dynamic recovery in ferrite and austenite.
In general, the paper shows a well-described modeled analysis and experimental validation, with some gaps that have been addressed in the review. Some additional observations should be taken into account to improve the paper.
- The title does not reflect the microalloyed steel material used in the study. Include the specific material in the title of the paper and not just “steels”.
I will describe it as “continuously cast microalloyed steels”, as the material has not commercial name but internally designation.
- Include in the abstract the microalloyed steel material used in the study.
The material is a continuously cast microalloyed steel with an internal designation but no commercial name. In previous studies this material has been reported by steel A (https://www.mdpi.com/2075-4701/12/11/1808) and steel number 1 (https://doi.org/10.37904/metal.2021.4140).
- The introduction section should be improved by including an extensive analysis of the state of the art available in literature. This section shows a lack of analysis of the state of the art.
Thank you for your comment. I modified the introduction (lines 60-101). I included the work of other group with respect to ferrite films and ductility, and explained the aim of our research in a better way (above Figure 1).
Submission Date
22 August 2024
Date of this review
03 Sep 2024 19:39:15